# Moving Away from 12:12; the Effect of Different Photoperiods on Biomass Yield and Cannabinoids in Medicinal Cannabis

**DOI:** 10.3390/plants12051061

**Published:** 2023-02-27

**Authors:** Tyson James Peterswald, Jos Cornelis Mieog, Razlin Azman Halimi, Nelson Joel Magner, Amy Trebilco, Tobias Kretzschmar, Sarah Jane Purdy

**Affiliations:** 1New South Wales Department of Primary Industries, 105 Prince Street, Orange, NSW 2800, Australia; 2Southern Cross Plant Science, Southern Cross University, Military Rd., East Lismore, NSW 2480, Australia; 3School of Agriculture and Food, Faculty of Science, The University of Melbourne, Parkville, VIC 3010, Australia

**Keywords:** medicinal cannabis, photoperiod, cannabinoids, flowering

## Abstract

The standard practice to initiate flowering in medicinal cannabis involves reducing the photoperiod from a long-day period to an equal duration cycle of 12 h light (12L)/12 h dark (12D). This method reflects the short-day flowering dependence of many cannabis varieties but may not be optimal for all. We sought to identify the effect of nine different flowering photoperiod treatments on the biomass yield and cannabinoid concentration of three medicinal cannabis varieties. The first, “Cannatonic”, was a high cannabidiol (CBD)-accumulating line, whereas the other two, “Northern Lights” and “Hindu Kush”, were high Δ9-tetrahydrocannabinol (THC) accumulators. The nine treatments tested, following 18 days under 18 h light/6 h dark following cloning and propagation included a standard 12L:12D period, a shortened period of 10L:14D, and a lengthened period of 14L:10D. The other six treatments started in one of the aforementioned and then 28 days later (mid-way through flowering) were switched to one of the other treatments, thus causing either an increase of 2 or 4 h, or a decrease of 2 or 4 h. Measured parameters included the timing of reproductive development; the dry weight flower yield; and the % dry weight of the main target cannabinoids, CBD and THC, from which the total g cannabinoid per plant was calculated. Flower biomass yields were highest for all lines when treatments started with 14L:10D; however, in the two THC lines, a static 14L:10D photoperiod caused a significant decline in THC concentration. Conversely, in Cannatonic, all treatments starting with 14L:10D led to a significant increase in the CBD concentration, which led to a 50–100% increase in total CBD yield. The results show that the assumption that a 12L:12D photoperiod is optimal for all lines is incorrect as, in some lines, yields can be greatly increased by a lengthened light period during flowering.

## 1. Introduction

*Cannabis sativa* is an economically and socially significant plant species with uses ranging from producing fibre for clothing; seed for animal and human nutrition; and psychoactive compounds for medicinal, religious, and recreational use.

Medicinal cannabis produces large quantities of cannabinoids and other diverse secondary metabolites in the glandular trichomes predominantly found on the female reproductive organs [1,2]. At least 113 cannabinoids and over 120 terpenes have been identified from the resin produced by the trichomes [3]. However, the two predominant target cannabinoids from medicinal cannabis remain: Δ9-tetrahydrocannabinolic acid (THCA) and cannabidiolic acid (CBDA) [2]. When the acid forms of these secondary metabolites are decarboxylated to their neutral forms Δ9-tetrahydrocannabinol (THC) and cannabidiol (CBD), respectively, they can interact with the mammalian endocannabinoid system for the treatment of non-communicable illnesses including sleep disorders, multiple sclerosis, appetite stimulation, and epilepsy [4,5,6].

Cannabinoid composition and abundance are largely under genetic control [7]. However, it is also recognised that environmental conditions and plant management strategies, such as growth substrate, water restriction, light spectra, and plant architecture training, may influence cannabinoid accumulation [8,9,10,11]. Thus, growers can optimise these factors to push for increased yields [12]. Photoperiod-sensitive crops, such as cannabis, align their development with the amount and timing of light that is available, and in controlled environments require specific lighting schedules to maintain or switch developmental stages. To maintain the vegetative state, growers typically use daily photoperiods of ≥16 h of light (≤8 h dark), and when flowering is to be initiated, the photoperiod is abruptly switched to 12 h of light and 12 h of darkness [12,13,14]. The 12 h light/12 h dark (12L:12D) rule reflects the fact that most cannabis genotypes are photoperiod sensitive and exhibit short-day flowering dependence. However, the native distribution for Cannabis is thought to extend from Siberia to China and now extends worldwide, including through much of the southern hemisphere [15,16]. In a phylogeographic study of an extensive collection of hemp germplasm in China, it was observed that the accessions fell into three distinct haplogroups that were differentiated by adaption to a high, middle, or low latitudinal gradient [15]. The authors studied the climatic factors that best correlated with the haplotype distribution pattern and found the strongest relationship was with mean daylength, accounting for ≈60% of the observed variation [15].

Differences in the critical photoperiod between accessions were reported in a comprehensive study that included 15 medicinal (“essential oil”) varieties [17]. It was reported that 14 genotypes had a critical photoperiod requirement of ≥14 h, with 3 genotypes initiating flowering in photoperiods ≥15 h [17]. The initiation of flowering in response to photoperiod is also recognised to vary with cultivar and environment in industrial hemp [17,18]. A U.S. company, The Hemp Mine, provides the photoperiod required to initiate flowering for five genotype which range from 13 h:55 m to 13 h:18 m [19]. Although cannabis is unequivocally considered a short-day flowering plant, the evidence from these studies show that flowering is frequently initiated in a photoperiod in excess of 12 h, i.e., a slightly longer day than night. The differences in hemp flowering time have been linked to their origin with cultivars adapted to more northern climates (in the northern hemisphere), showing reduced photoperiod sensitivity (and thus earlier flowering) than southern adapted cultivars grown under the same photoperiod [18]. These previous observations support the question as to whether the blanket rule of 12:12, applied by commercial glasshouse/indoor growers, is actually optimal for maximum yields for all cultivars.

The cannabinoid profile has also been reported to change in response to extended photoperiods. In an experiment on hemp where the plants were exposed to an extra two hours of light, a significant increase in Δ9–THC was observed, nearly doubling it from its control level, while cannabidivarin (CBDV) levels were reduced by 50% [20]. The timing of floral initiation is also important for the medicinal grower as yield is expressed as g cannabinoid m^−2^ day^−1^ [21]; therefore, any additional days to maturity lowers yield over time. In a study into the flowering time of cannabis explants, it was observed that photoperiods in excess of 13.2 h extended the number of days to flowering, with the shortest period of 12 h light causing the most rapid flowering [14]. Under natural field conditions, photoperiods progressively change with the seasons, whereas in controlled facilities, the changes are generally abrupt. Whether a progressively changing photoperiod, compared to a single sudden change, affects yield attributes in a glasshouse/indoor system remains to be determined.

The genetic control of short-day (SD) flowering is less characterised than in long-day flowering plants (e.g., Arabidopsis), and most of the knowledge in SD plants comes from rice (a monocot). However, the main mechanism for photoperiodism in long-day (LD) plants appears to be conserved in SD plants. The principal flowering-control gene in SD rice is HD3a, which is homologous to the main control gene, FT “Florigen”, in LD plants [22]. In LD flowering, FT is activated by the CONSTANS (CO) protein, which then initiates flowering [23]. A SD homolog of CO, Hd1, has been identified in rice, and it regulates the expression of HD3a [24]. Both CO and Hd1 are regulated by the circadian clock and show a diurnal pattern of oscillation. The periodicity of the expression pattern does not change with daylength, which means that the peak of expression occurs at different times of the day depending upon the season [25]. When the expression of Hd1 occurs in the dark the protein is stabilised and triggers the expression of HD3a [25]. As different cannabis lines were cultivated in new latitudes, variation in response to photoperiod would have also been selected for, as plants that flowered to late, or too early, would have been poor performers. Despite this likelihood, to our knowledge, the effect on yield of changes to the flowering photoperiod away from a static 12L:12D, e.g., 14L:10D, or a progressive shortening or lengthening, remains to be studied for medicinal lines.

We sought to test the effect of either lengthened or shortened photoperiod away from 12L:12D on flowering time, biomass flower yield, and target cannabinoid concentrations in three medicinal cannabis varieties. Nine treatments were tested that utilised either a static lengthened or shortened period, or treatments that lengthened or shortened over the flowering periods in two stages.

## 2. Results

### 2.1. Reproductive Development

All plants in all three treatments had initiated flowering, as measured by the presence of pistils, on or before DAC 46 when the transfer between treatments took place (Table 1). While the high CBDA accumulating line, Cannatonic, was the latest to start developing pistils (in all treatments), it was the quickest to develop floral trichomes, being the first to reach 100% of plants with visible trichomes in all three treatments. The shortened photoperiod of 10L:14D tended to be the slowest treatment in which pistils and trichomes were observed. Conversely, the lengthened period of 14L:10D was the only treatment in which some plants of one line had produced trichomes at DAC 34 (Table 1). Overall, these results showed that the initiation of flowering was delayed by reducing the photoperiod to 10L compared to 12L and 14L photoperiods for all three lines.

### 2.2. Plant Height

Plant height was measured weekly, and all plants in the 10L and 12L treatments had started to plateau in height by DAC 40, whereas the 14L plants kept increasing in height until ≈DAC 46 and were taller than the other treatments (Figure 1). Only minimal increases in height were observed in plants transferred from shorter to longer daylengths.

### 2.3. Flower Biomass Yield

The flower biomasses (Flower DW g Plant^−1^) were significantly higher for the 14L treatment (Figure 2) than for the 12L control for all three lines. In Cannatonic and Northern Lights, 14L > 12L and 14L > 10L were also significantly higher than 12L. These were the only significant differences, but in Cannatonic and Hindu Kush, the three treatments that resulted in the shortest photoperiods, 10L, 10L > 12L and 12L > 10L, also trended to result in the lowest yields.

### 2.4. Cannabinoid Concentration

In Cannatonic, the CBD concentration ranged from 7 to 11%, depending on the treatment (Figure 3). The highest levels were observed in the treatments that started at 14L (i.e 14L, 14 < 12L, 14 < 10). In treatments that started at 10L, the % CBD was significantly lower compared to 12L, even if they ended with 14L (Figure 3). This indicated that the CBD concentration was determined in the first 28 days in this genotype and that an extended photoperiod increased production. In contrast to Cannatonic, the 14L treatment significantly reduced the % THC by around 2/5th from the standard 12L treatment in Northern Lights (12% down to 7%) and in Hindu Kush (10% down to 6%) (Figure 3). No significant differences between any of the other treatments were found for Northern Lights and Hindu Kush (Figure 3). The treatments that started at 14L and then dropped to 12L or 10L produced the same %THC as the 12L treatments. In Northern Lights, treatments that had the same total units of light but applied at different times such as 12L < 10L and 10L > 12L, and 14L < 10L and 10L > 14L had the same concentration of THC. This demonstrated that the timing of the shorter photoperiod, whether it was earlier or later in flowering, did not affect cannabinoid concentration. Only if the extended 14L period was maintained throughout the experiment was cannabinoid concentration decreased. The 12L photoperiod was optimal for THC production in both Northern Lights and Hindu Kush (Figure 3).

The relationship between biomass yield and cannabinoid concentration for the three static treatments (10L, 12L, and 14L) is shown in Figure 4. However, the critical point at which flower biomass started to decline was not identified by this study as, in all lines, the DW yield increased between 12 and 14L. These data also suggested that the optimal photoperiod for yield in Cannatonic may be longer than 14L as both the biomass and cannabinoid concentration continued to increase up to 14L (Figure 4).

### 2.5. Physical Appearance of Flowers

Photographs of the flowers from each line from the three static treatments (10L, 12L, and 14L) are shown in Figure 5. Only the 14L treatment was notably different from the other two treatments. In the 14L treatment in Cannatonic, there was less/no visible anthocyanin accumulation and instead of a single inflorescence at the top of the stem, there were multiple inflorescences. In both Northern Lights and Hindu Kush, the 14L inflorescence (“sugar”) leaves were longer than in the other treatments, and in Hindu Kush, also wider. The trichome density was visibly less in Northern Lights and Hindu Kush in the 14L treatment, and pistil senescence appeared delayed (Figure 5). In the replication of this experiment, all nine treatments were photographed weekly, and the final photographs taken at DAC 74 are shown in Appendix A. The same trends in the static treatments were observed in the replication. In Cannatonic, plants that had started in 14L and then were moved to shorter photoperiods had larger, clustered flowers, as seen in the 14L treatment in Figure 5, but the pistils were more senesced. In Northern Lights and Hindu Kush, the plants that started in 14L and then moved were visibly less senesced, but the trichome density appeared higher than the 14L treatment (Figure 5, Appendix A).

### 2.6. Total Yield

In Cannatonic, the combination of increased biomass and increased CBD concentrations resulted in a doubling of yield in the 14L treatment compared to the 12L control (Figure 6). Yields of CBD were also nearly doubled in 14L > 12L and 14L > 10L.

In Northern Lights, the yield of THC was increased in the 14L > 12L compared to 12L, because of the increase in biomass coupled with no change in the THC% (Figure 6). No significant difference was observed in Hindu Kush (Figure 6). Although there was no statistically significant negative effect of the 10L treatments on total cannabinoid yields, there was a trend for reduced yield in the 10L treatment in both Cannatonic and Hindu Kush. A pair of two-way t-tests (assuming unequal variances) to compare 10L and 12L in Cannatonic and Hindu Kush showed a significant difference (*p* ≤ 0.05) between the two photoperiods for both lines (Figure 6).

## 3. Discussion

The practice of initiating flowering through a reduction in photoperiod to 12L:12D is long-held standard methodology in cannabis production, as demonstrated by its reference in early cannabis growing guides [26], as well as modern publications since legalisation [12,13,27,28]. The fact that this blanket rule is assumed to be optimal for all varieties is quite remarkable considering the diverse latitudinal origins of cannabis [16] and known variation in photoperiod-dependent and -independent (a.k.a autoflower trait) flowering time control in cannabis.

Our results showed differences between three varieties in their response to different light treatments with some significant yield increases in response to 14L. Most significantly, the high-CBD line (Cannatonic) showed cannabinoid yield increased to more than double when an early 14L photoperiod was applied compared to the standard 12L. In contrast, one of the two high-THC lines tested (Northern Lights) only showed a 50% yield increase in the 14L < 12L treatment, whereas the second high-THC line tested (Hindu Kush) did not show any significant yield effects.

All three varieties showed a positive response to 14L in the early flowering phase with regard to height and flower biomass, which can be explained as a result of the extra energy available for photosynthesis. For two varieties (Cannatonic and Hindu Kush), the 14L treatment tended to lead to more flower biomass than 14 > 12 or 14 > 10, indicating that the reduction in energy later in the flowering phase negatively affected flower initiation and/or development. The absence of this effect for Northern Lights is possibly a result of earlier flower maturation in this variety, with the more-developed inflorescences less affected by the reduction in available light energy. Alternatively, stored carbohydrate reserves could have been remobilised closer to harvest, acting as a buffer against the reducing photoperiods in the second half of flowering.

In terms of cannabinoid concentration, Cannatonic responded very differently to the photoperiod treatments compared to the two high-THC varieties. The cannabinoid concentration of Cannatonic appeared to be positively correlated with the photoperiod length in the first flowering phase with no obvious differences resulting from the rotation of plants between treatments at DAC 46. Thus, for this variety, the cannabinoid concentration appears to be mostly determined in the early flowering phase, with more energy available leading to more cannabinoid accumulation. This may indicate that Cannatonic accumulates carbon reserves early in the flowering phase, which are mobilised for trichome production and/or cannabinoid biosynthesis later. As the floral tissues were still mostly forming at this stage, it is unlikely reserves were accumulated there. Instead, sugars could be remobilised from starch reserves accumulated in the stems and/or roots. In citrus, flowering is the most carbon-demanding developmental phase (greater than fruiting), and it costs ≈90 mg glucose per flower from flower development to anthesis [29,30]. The source of this carbon is largely from the remobilisation of starch reserves, of which 42%, equating to 230 g of non-structural carbohydrate (NSC), originated from the roots [30]. Citrus trees are perennial and so have a larger, woodier root system than an annual such as cannabis; however, in cotton, the remobilisation of carbohydrates from both roots and stems during flowering has also been documented. Furthermore, the efficiency of remobilisation was also greater from the cotton roots rather than the above-ground biomass, but the absolute units of carbohydrate remobilised were greater from the stems [31]. To our knowledge the carbohydrate dynamics of a flowering medicinal cannabis plant have not been studied, and thus the potential source of remobilised carbohydrate remains to be characterised.

In contrast, for the two THC varieties, not only was the above-mentioned positive correlation absent, a clear penalty for 14L was visible which was not apparent for Cannatonic. The inflorescences of the 14L treatments of Northern Lights and Hindu Kush showed sparser trichome distribution and elongated leaf shape, indicating that the reproductive phase may not have reached completion in this treatment, likely because of either a later transition from vegetative adult to reproductive or a slower ripening in the reproductive phase. It is known that the timing, density, and distribution of trichomes is related to developmental phase [32]. In maize, trichomes are confined to the leaf margins during the juvenile phase but also appear on the upper leaf-surface of adult leaves [32]. In juvenile Arabidopsis leaves, the trichomes are absent from the abaxial (lower leaf) surface but are present on both faces once the adult phase has been reached; furthermore, the leaf shape changes from flat and round/orbicular when juvenile to curled, serrated, and spatulate when mature [32,33]. The trichomes on Arabidopsis are different from those of cannabis because they are non-glandular, but transcription factors for trichome development from Arabidopsis have also been identified in cucumber plants, which possess both glandular and non-glandular trichomes, demonstrating conservation in the regulatory pathways [15]. Chien and Sussex [33] observed that both flowering and trichome development were photoperiod sensitive in Arabidopsis, but the timing of the sensitive phase differed, with trichome initiation being earlier than flowering. From this, it was concluded that although the photoperiod control of flowering and trichome development may be regulated by the same mechanisms, the timing of the events is separable [33]. This appears to be the case in our study where higher flower biomass yields were obtained in all lines under the 14L photoperiod, but trichome density and productivity were negatively affected in the two high-THC lines.

The significant gains in total CBD yield in treatments starting with 14L for Cannatonic resulted from increases in both flower biomass and flower CBD concentration (Figure 4), showing that this variety was able to utilise the extra energy available both in early and late flowering for growth and flower/trichome development.

Although the two high-THC lines showed similar increases in growth and flower biomass with increased photoperiod, a strong decrease in cannabinoid concentration eventuated once the photoperiod was extended beyond 12L. As a result, Hindu Kush showed no significant differences in total cannabinoid yield between treatments as the increase in flower biomass (39%) almost perfectly matched the decline in THC concentration (40%) when comparing 14L to 12L. For Northern Lights, gains were less than for Cannatonic but still significant, resulting from the 14L < 12L treatment not having a flower biomass reduction compared to 14L or a cannabinoid concentration reduction compared to 12L, leading to a net gain of around 50%. For Cannatonic, it is possible that extending the photoperiod even further may also further increase yield, although it is more likely that maturation issues similar to those seen in the high-THC varieties will limit further gain.

The different responses of the three lines to photoperiod duration probably reflects the geographical origin of their genetics. In a study of 654 cannabis genotypes, it was observed that those adapted to more northerly latitudes flowered earlier than southern adapted lines when grown in a more southerly location (with shorter daylengths), demonstrating greater photoperiodic sensitivity [15]. Flowering time in high-cannabinoid hemp cultivars was observed to show a pattern of variation that was consistent with control by several major-effect loci [34]. One of the major loci that was alluded to in this study was subsequently identified and named “*Autoflower1*” in a population produced from the cross between a photoperiod-sensitive and -insensitive (“autoflowering”) cultivar [34,35]. *Autoflower1* is a recessive trait and homozygous plants flowered under continuous light, whereas those that were heterozygous would not flower under continuous light and flowered 2 weeks later under field conditions [35]. This study also identified a second locus “*Early1*” which conferred an earlier flowering time of 2–4 weeks in one cultivar [35]. A candidate gene for *Early1* is *HD16/Early flowering1*, which is a major flowering time gene in rice [35,36]. Allelic variants of a related major flowering time gene, *HD1*, were shown to strongly correlate with flowering time in rice, with nine allele types (nucleotide polymorphisms) accounting for 50–60% of the variation. It is likely that the range in response to photoperiod observed in cannabis, including those used in this study, is the result of allelic variation of major flowering genes such as those underlying *Autoflower1* and *Early1*.

Medicinal cannabis products are usually described by their % cannabinoid concentration, and there has been a trend for increasing levels since the 1980s [37]. In a study into potency and sales in Washington State, 90% of sales were for flowers >15% THC, whereas sales for those <10% accounted for only 2% in 2014–2016 [38]. Therefore, for inhalable products with high concentration, the photoperiod needs to be optimised to maximise cannabinoid concentration per inflorescence. However, pain relief is one of the main conditions for which medicinal Cannabis is prescribed, and a number of studies have demonstrated that a lower concentration of 5–10% THC is effective with minimal side effects [39,40,41]. If a lower-than-maximum level from a particular variety was desirable, our results indicate that this could be achieved by lengthening the photoperiod past 12L in order to achieve increased inflorescence yields with reduced cannabinoid concentration. Electricity is a major cultivation expense, as it estimated to require Australian growers 4kWh of energy to produce 1g of dried flower per m^2^ [42]. The environmental impacts for indoor cultivation are also considerable; in the USA, it is reported that the greenhouse gas emissions range from 2.283 to 5.184 kg CO_2_ equivalent to produce 1kg of flower [43]. Therefore, achieving gains without extending the duration (or better still, shortening) of the artificial lighting period would be preferred. Our results showed an almost doubling in yield for Cannatonic when the lighting was set to 14L for the first 28 days of flowering and then 10L for the remaining 29 days, which equates to the same total number of lighting hours as a 12L:12D cycle. This finding illustrates that a progressively reducing photoperiod can have substantial impacts on yield for certain specific genetic backgrounds with no additional inputs required.

In summary, our results showed that distinct varieties can exhibit markedly different responses to changes in photoperiod length, and the standard photoperiod for the flowering phase of 12L:12D is not optimal for all varieties. In particular, cannabinoid yields (g cannabinoid plant^−1^) can be more than doubled by increasing the photoperiod during the flowering phase from 12 h to 14 h, as demonstrated by the Cannatonic line, with the increase in cannabinoid yields driven by gains in both flower biomass and flower cannabinoid concentration. A 14L > 10L photoperiod also achieved a strong yield benefit which utilises the same number of light hours as 12L and therefore incurs no extra electricity costs. For one high-THC line, a 14L > 12L photoperiod increased THC yields by 49%, driven by a gain in biomass only (no change in % THC), whereas a second high-THC line did not show any significant differences. As this treatment also benefitted Cannatonic, this may be the best “all-round” treatment optimal for mixed cultivation and untested varieties.

## 4. Materials and Methods

### 4.1. Plant Material and Growing Conditions

Three medicinal cannabis genotypes were utilised that had been supplied from Cann Group Ltd (https://www.canngrouplimited.com/, accessed on 17 January 2023). They comprised of one high-CBDA line “Cannatonic” and two high-THCA lines, “Hindu Kush” and “Northern Lights” (previously referred to as “CBD1”, “THC6”, and “THC1”, respectively [21]).

All plants were cultivated in an Australian Government Department of Health and Aged Care Office of Drug Control (ODC) approved secure facility. All experiments were conducted under a Commonwealth license and associated permits. The temperature was maintained at 25 °C and humidity at 50%. Plants were grown in controlled environments (CE) throughout their life cycle (details below). Throughout the flowering period, plants were moved around within a CE, and three times over the course of the experiment, plants were transferred to a different CE and the conditions were re-set for that treatment. The purpose of these movements was to control for any environmental differences between the CEs. The entire experiment was repeated twice to ensure replicability of the findings.

The cloning and propagation method used has been previously described [21]. Experimental plants were cloned from donor mothers. New growth stems of approximately 15 cm were excised from the mother. All leaves up the sides of the stem were removed, leaving the top leaf bunch. The bottom of the stem was then cut diagonally across a node using a scalpel in order to form a clone approximately 12 cm in height. The top leaf bunch was trimmed to the height of the smallest emerging leaf to reduce water loss and prevent the clones from overlapping in the propagation dome. The bottom 1 cm of the stem, from which the roots would form, were lightly scraped with a scalpel and then dipped in hormone gel (Clonex Purple, Yates, DuluxGroup, Clayton, Australia) and placed in an organic propagation cube (Eazyplug CT12, Goirle, The Netherlands, eazyplug.nl). Once the propagation tray was full of new clones, it was placed in a propagation dome (Smart Garden heavy duty 3-piece propagation kit, Epping Hydroponics) for 18 days under an 18 h light/6 h dark (18L:6D) photoperiod in a growth cabinet (Conviron A2000, Conviron Asia Pacific Pty Ltd., Grovedale, Australia) at a light intensity of 100 µmol m^2^ s^1^ and a temperature of 25 °C. Humidity monitors were placed in a dome, and the humidity was progressively reduced over the fourteen-day propagation period. Plants with established roots were then potted into 1.8L pots containing a 30:70% blend of perlite and coco-coir, electrical conductivity (EC) = <0.5 mS/cm (Professors Nutrients, Truganina, Australia). Plants were then transferred to their flowering controlled environments under a PAR of 700 µmol m^2^ s (Viperspectra PAR 700, Viparspectra, Richmond, VA, USA) https://viparspectraled.com.au/ accessed on 17 January 2023). The temperature was maintained at 25 °C, and blackout curtains removed the risk of light leakage. Plants were watered and fed using a commercial fertigation recipe, E.C. = 2.2 mS/cm and pH = 6.

### 4.2. Treatments

Each treatment contained 15 replicates of each of the three genotypes in a fully randomised design. The three zones were programmed to one of each of the following photoperiods (hours light/hours dark): 10L:14D, 12L:12D, 14L:10D. All plants were maintained in these treatments for 28 days, which was the day after cloning (DAC) 46, the half-way point of the flowering treatments. At DAC 46, 10 plants from each of the three treatments (initially containing 15 plants) changed treatment so that 5 experienced a lengthening, and 5 a shortening, of the photoperiod, resulting in 9 combinations of 5 plants from each genotype. Plants then stayed in these treatments for the remaining 29 days of flowering before harvest on DAC 75. A description of the treatments is provided in Table 2. The direction of the symbol in the treatment name indicates whether the photoperiod increased (>) or decreased (<) in duration after DAC 46.

### 4.3. Measurements

The development of flowering was scored fortnightly until DAC 34 and then weekly thereafter for the presence or absence of pistils (1/0) and trichomes (1/0). Height was measured weekly.

Harvesting: The harvest took place on DAC 75. Plants were excised at the base, and then the whole plant was weighed (whole plant FW). The large fan leaves were removed, and the flowers were manually stripped from the stem and trimmed using a mechanical trimmer (TrimPro ROTOR, Saint-Jean-sur-Richelieu, QC, Canada). The trimmed flowers were re-weighed (flower fresh weight) and placed into a foil tray. The flowers were dried in a dedicated drying room at 21 °C and 50% humidity until no further reduction in weight was observed (9 days). The samples were then re-weighed, and the total flower dry weight (g plant^−1^) was calculated.

### 4.4. Analytics

The method used for the quantification of cannabinoids has been previously described [44]. Five biological replicates for the three genotypes in each of the nine treatments and controls were analysed for THCA and THC (Hindu Kush and Norther Lights) and CBDA and CBD (Cannatonic). Total THC and CBD was then calculated with the following formulae: Total THC = THC + (THCA × 0.877) and Total CBD = CBD + (CBDA × 0.877).

Three florets were randomly removed from the dried subsample flower material from each individual plant and ground to fine powder in liquid nitrogen. A 0.1 g subsample was used for cannabinoid quantification in 100% ethanol using sonication (SONICLEAN, Soniclean®, Dudley Park, Australia) at 50/60 Hz for 30 min, followed by centrifuge at 10,000 rpm for 10 min. The ethanolic extracts were stored at −10 °C until use. The ethanolic extracts were diluted and analysed using high performance liquid chromatography–quadrupole time of flight mass spectrometry (UHPLC-QToFMS, Agilent Technologies, Santa Clara, CA, USA). Separation was achieved using a reversed-phase column (Agilent Infinity Poroshell 120, HPH-C18, 2.1 × 150 mm, 2.7 µm, narrow bore LC column, Agilent Technologies, Santa Clara, CA, USA) and methanol–water–acetonitrile and acetonitronitrile mobile phases, each containing 0.1% formic acid (*v*/*v*). Analysis was performed using Quant Analysis Software 10.2 (Agilent Technologies, Santa Clara, CA, USA). Cannabinoid peaks were identified according to their m/z values and retention times by calibration against cannabinoid standards (Novachem, VIC, Australia).

The total yield (mg plant^−1^) of each cannabinoid was calculated as
((%Cannabinoid/100) × Total flower DW g) × 1000 = mg cannabinoid plant^−1^
(1)

### 4.5. Graphics and Statistics

All graphics and statistical analyses were performed in R 3.1 [45]. Multiple comparisons were performed with Dunnet’s tests [46] in order to identify significant differences between the 8 treatments and the control (12L). Individual t-tests were performed for pair-wise comparisons.

## Figures and Tables

**Figure 1 plants-12-01061-f001:**
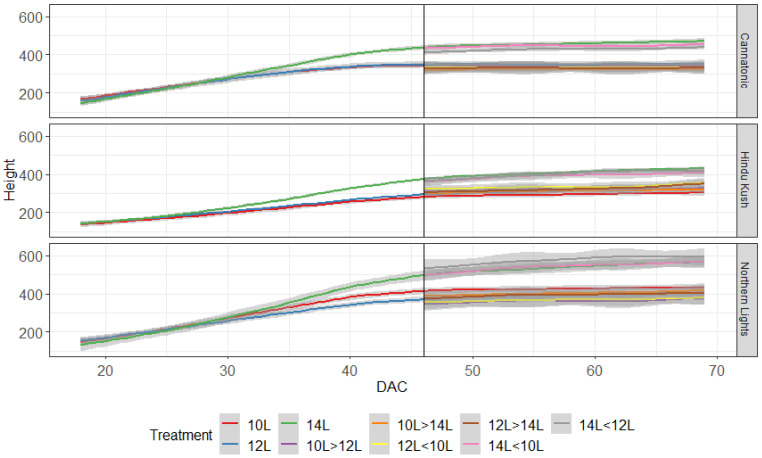
Plant height (mm) for the 9 treatments over time. The graph starts when plants entered the flowering treatments. The black line shows the point at which plants moved to their second treatment. DAC 46 N = 15, after DAC 46 N = 4−5.

**Figure 2 plants-12-01061-f002:**
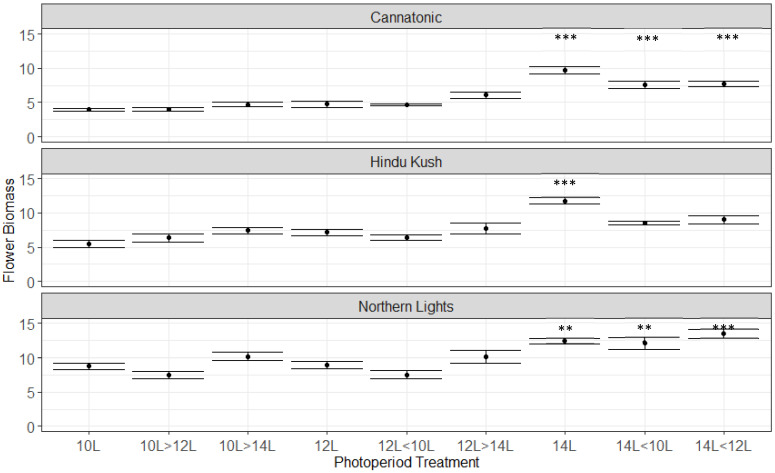
Flower yields (g DW Plant^−1^) for the 9 treatments. Asterisks above the error bars indicate that the treatment was significantly different to the control 12L treatment, Dunnet’s test, *** *p* ≤ 0.001, ** *p* ≤ 0.01, N = 4−5 +/− SE.

**Figure 3 plants-12-01061-f003:**
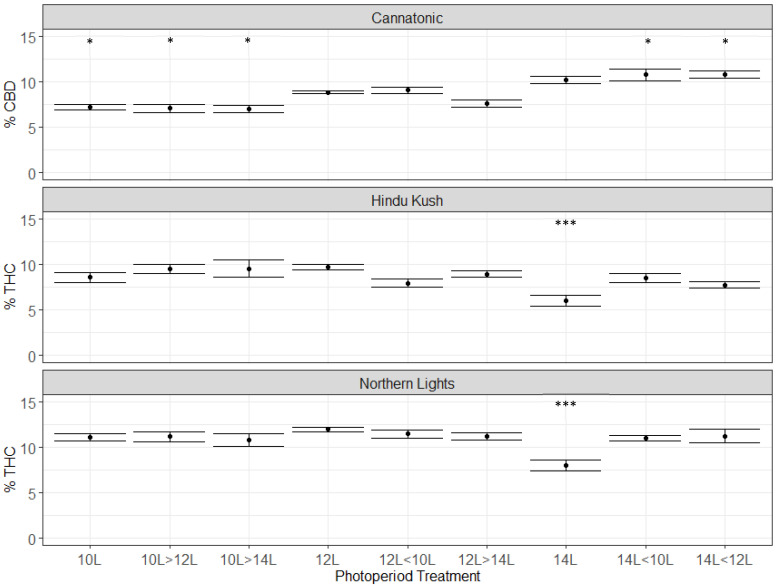
Cannabinoid data from the three genotypes: Cannatonic, Northern Lights, and Hindu Kush. Asterisks above the error bars indicate that the treatment was significantly different to the control 12L treatment, Dunnet’s test, *** *p* ≤ 0.001, * *p* ≤ 0.05, N = 4−5 +/− SE.

**Figure 4 plants-12-01061-f004:**
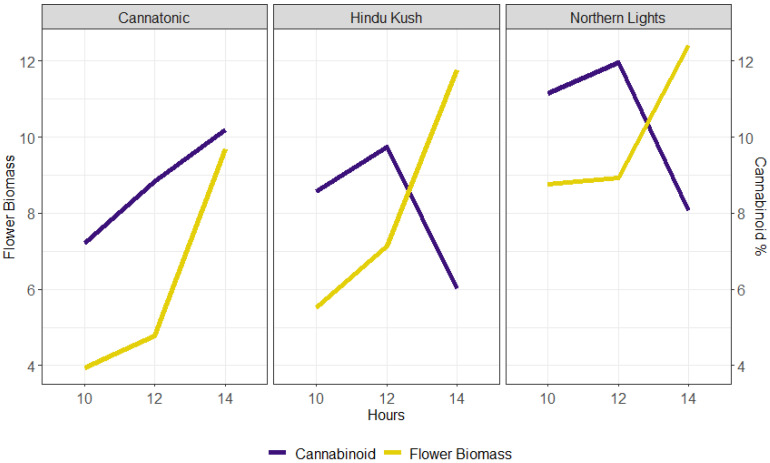
Relationship between flower biomass yield (g DW plant^−1^) (purple line) and the % target cannabinoid (Cannatonic = CBD and Northern Lights and Hindu Kush = THC) (yellow line) in the three static photoperiods.

**Figure 5 plants-12-01061-f005:**
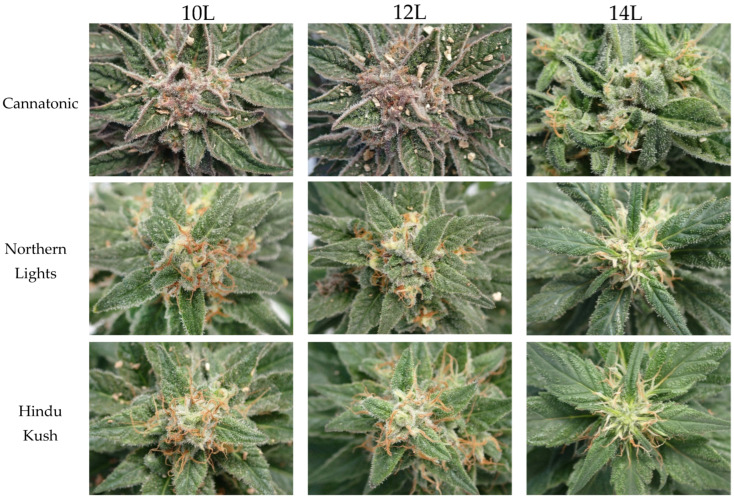
Photographs of the top of the main flowering stem taken at DAC 67 from the three static treatments.

**Figure 6 plants-12-01061-f006:**
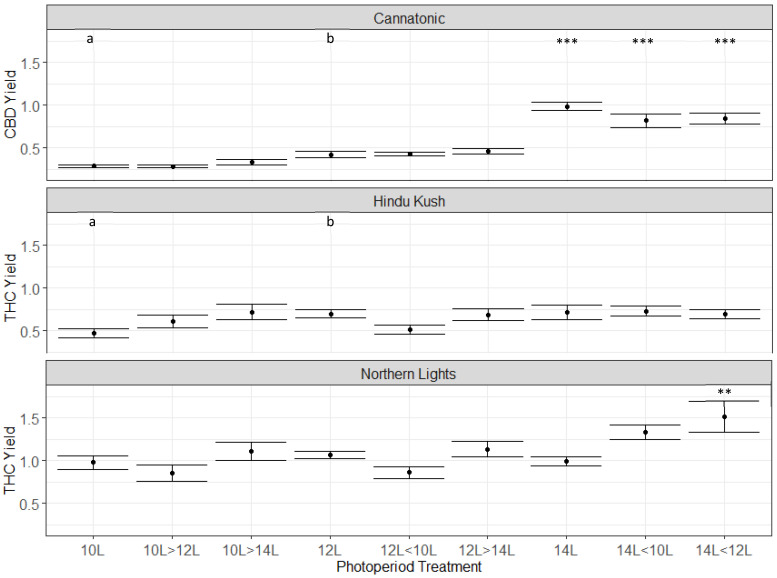
Total yield (g plant^−1^) of the target cannabinoid in the three genotypes. Asterisks above the error bars indicate that the treatment was significantly different to the control 12L treatment, Dunnet’s test, *** *p* ≤ 0.001, ** *p* ≤ 0.01, N = 4−5 +/− SE. Different letters above the 10L and 12L treatments show significant differences between those two treatments (two-way t-test assuming unequal variance) *p* ≤ 0.05.

**Table 1 plants-12-01061-t001:** Reproductive development in the three starting photoperiods. The % of plants showing the presence of pistils and trichomes is shown for difference days after cloning (DAC). N = 15.

		% of Plants with Pistils		% Plants with Trichomes			
Genotype	Photoperiod	DAC 25	DAC_34	DAC_25	DAC_34	DAC_41	DAC_46
Cannatonic	10L	0	100	0	0	100	100
	12L	0	100	0	100	100	100
	14L	0	100	0	80	100	100

Northern Lights	10L	60	100	0	0	26.6	100
	12L	66.6	100	0	40	100	100
	14L	73.3	100	0	33.3	100	100

Hindu Kush	10L	0	100	0	0	26.6	100
	12L	20	100	0	0	100	100
	14L	20	100	0	6.6	100	100

**Table 2 plants-12-01061-t002:** Description of the treatments applied. Five replicates per genotype were included for each treatment.

Treatment 1 (Light/Dark)	Treatment 2 (Light/Dark)	Name	Description
10L:14D	10L:14D	10L	10L:14D for duration of experiment
10L:14D	12L:12D	10L > 12L	Increased photoperiod from 10L to 12L (+2 h light)
10L:14D	14L:10D	10L > 14L	Increased photoperiod from 10L to 14L (+4 h light)
12L:12D	12L:12D	12L	12L:12D for duration of experiment
12L:12D	10L:14D	12L < 10L	Decreased photoperiod from 12L to 10L (−2 h light)
12L:12D	14L:10D	12L > 14L	Increased photoperiod from 12L to 14L (+2 h light)
14L:10D	14L:10D	14L	14L:10D for duration of experiment
14L:10D	10L:14D	14L < 10L	Decreased photoperiod from 14L to 10L (−4 h light)
14L:10D	12L:12D	14L < 12L	Decreased photoperiod from 14L to 12L (−2 h light)

## Data Availability

Data are contained within the article and Appendix A.

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
