# Peer review of "Moving Away from 12:12; the Effect of Different Photoperiods on Biomass Yield and Cannabinoids in Medicinal Cannabis"

_plants, 2023, doi:10.3390/plants12051061_

Round 1
Reviewer 1 Report
Report on plants-2189056-peer-review-v1
In the main this is an interesting, well-written manuscript, with results and discussions that should be made available to the readership of the journal ‘Plants’.
There are a number of issues, one in particular that must be addressed to avoid confusion over the use of terms ‘content’ and concentration.
The authors write of content when indeed they are referring to ‘concentration, for content = [concentration X mass]. For example, in lines 27 and 28, concentration should replace content, and in many other instances. I don’t believe ‘content should be used at all, for the authors use ‘yield’ of THC or CBD.
Also, the use of abbreviations for the first time must be qualified by the full term, for example in line 18 it should read …a high cannabidiol (CBD) accumulating… and in line 19 , to read ..were high Δ9-tetrahydrocannabinol (THC) accumulators.
Line 19 should read The nine treatments tested, following 18 days under 18 h light 6 h dark following cloning and propagation, included a 19 standard 12L:12D period, a shortened period of 10L:14D and a lengthened period of 14L:10D.
Line 21 I cannot find evidence that this shift was midway through flowering, so please add some detail.
Line 69 to read ‘provides’
Line 72 to read ‘have’
Line 73 earlier flowering under what condition(s)?
Line 78 I don’t believe this is the correct reference?
Line 83-85 rates of change of daylength are also important in governing some developmental processes.
Line 92 to read ‘regulates’
Line 103 to read ..lengthened or shortened photoperiod compared to 12L :12D, or treatments..
Line 123, 176 and elsewhere, always leave a space between a numeral and unit
Line 128 and 132 μmol m-2 s-1
Line 131 define CEs or use ‘growth cabinet’
Line 161 ‘weekly’ doesn’t match the time periods in Table 2.
Line 167 is % biomass used again. I don’t see it mentioned.
Line 176 .. in liquid nitrogen.. A 0.1g subsample was used for cannabinoid quantification..
Line 181 2.7 μm
Line 189 Graphics and statistics
Line 229 The black vertical line at DAC 46 shows…
Line 232 and 233 remove capitals where not needed.
Line 282 start a new paragraph, and remove paragraph at line 286, so that reference to % THC in line 287 clearly refers to Northern Lights.
Line 329 … between 12L and 14L. These data also suggested that [data are always plural]
Lines 388-389 … in a doubling of CBD yield in the 14L treatment compared to the 12L control (Figure 6). Yields of CBD were also nearly..
Line 394 variances
Figure 6 verify the titles for the y-axes.
Line 443 response
Line 451, but think again about this, could it be a photoperiodic effect [i.e., separate from energy for photosynthesis], and that reallocation of NSC assimilates closer to harvest effect the noted differences?
Line 451 absence
Line 489 yields
Line 540 …with the increase in cannabinoid yields driven by gains in both 541 flower biomass and flower cannabinoid concentration.
Line 544 is this correct [14L>12L], and the other three instances?
Line 544 ..increased THC yields by…
Line 558 Data are contained..
In the References, take note of using italics for Latin names [e.g., 573..]and loci traits [e.g., 664], not using capitals in article titles, abbreviations of journal titles [588, 590,628, 665]
Line 634 BMC Plant Biol
Line 663 more detail required.
Reviewer 2 Report
The authors submitted a paper presenting the effect of different photoperiods (lengthening or shortening away from 12L:12D interval) on the yield of flower biomass and target cannabinoids in the three medicinal Cannabis varieties. The experiment is logically designed (controlled environments) and hemp varieties differring in their genetic background, geographical origin and thus also physiological properties were properly chosen based on previous authors´ experience (Purdy et al. 2022). The obtained results have confirmed the authors´ hypothesis, that moving away from 12L:12D may affect the behaviour of studied hemp cvs. (flowering timing, flower and trichomes development/abundance, leaf morphology as well as the production of substances of interest – CBD and THC). Nevertheless, the particular cvs. responded to the same treatment differently – this fact is correctly discussed based on relevant recent literature and comparison with plant models, e.g A. thaliana and rice (e.g. the role of some major-effect loci connected with photoperiodic induction of flowering). Finally, the authors concluded, that proper selection of hemp cvs. combined with suitable growing conditions may result in high flower biomass and cannabinoids yields with no extra electricity costs. The submitted study has provided some new theoretical and methodological data and may be of practical use for those dealing with medicinal Cannabis production.
The paper may be published in Plants journal after minor formal improvements (the References should be unified according to the format of Plants journal; some typos in the text).
Reviewer 3 Report
The authors used three genetically modified Cannabis plants to evaluate their response in different environmental conditions, to ensure a higher yield of natural compounds, with minimum consumption. The reviewer finds the experiments quite complex, with sufficient statistical considerations. As a suggestion, due to recent nomenclature, the authors might consider to change ‘secondary metabolites’ with ‘natural compounds’, throughout the text. Aside from small technical editing errors, which the reviewer would leave behind for now, since they might get corrected until the next review, few comments are written bellow:
1. Don’t you consider it might be better to change ‘donor mother’, to ‘donor parent’? I believe this could be the international S.I. for cloning terminologies.
2. This might be only for my case, but the treatment is slightly hard to follow. When did it actually begin? After 18 days of 18L:6D, after 46 days?
3. Why did you multiply with 0.877 when calculating Total THC and CBD?
4. Is it possible to render a better graphic for figure 1? I believe these results are important, but not so visible in the graph as it is.
Good luck!
Author Response
The reviewer has requested no further changes.
Thank you again for reviewing our paper.
Reviewer 4 Report
The manuscript, entitled 'Moving away from 12:12; the effect of different photoperiods 2 on biomass yield and cannabinoids in medicinal Cannabis', addresses issues related to optimising the photoperiod to increase the yield of desirable cannabinoids in different varieties of medicinal cannabis. This is a very up-to-date topic and I recommend its publication in Plants after taking into account the comments.
Row 75 leave one dot mark
Row 87 long-day, please add (LD)
Row 134 please provide the mineral content of the substrate , pH , EC salt concentration and the full composition of the nutrient solution for fertigation of plants(macro and micro nutrients what was the EC and pH )
Row 139 here it is necessary to explain in more detail that 10 plants from each of the three combinations where 15 plants were growing were changed the photoperiod, lengthening or shortening the photoperiod for 1/2 of them. Thus, nine combinations of 5 plants in each were obtained.
Row 226 the legend posted below the chart is not clear to me, the first three line markings and color should be for 10,12 and 14L and the next 6 should have other colors or markers that would clearly characterize the 2 types of changes,
Please make changes to the combination labeling on the chart
Row 305-318 please sign on each axis of the graph for one variety and the other % THC as it is signed for the Cannatonic variety %CBD
Row 404 what do the letters a and b mean on the chart
Row 404 please correct on axes signature g.Plant-1
If surveys were conducted twice, please provide the survey dates and are the results an average of the two study dates?Row 23 proszÄ™ wymienić wszystkie kombinacje ich skróty, nazwy np. jak w tabeli 1.
Row 336 the line on the graph is yellow, not orange
Row 274 please, either sign the axis of the graph the same as the title of the graph - flower yield or flower biomass and give the correct unit on the axis (g DW.plant-1) and do not repeat it in the title of the graph (note the same for figure1,2 and 4), or the unit always give in the title of the graph and on the axis give only the name of the studied attribute
Row 430 figure 6. I propose to leave only the abbreviation of the compound on the axis of the graph, i.e. THC, THC and CBD, and in the title of the graph write the unit - Total Yield (g.Plant-1), it will be more clearly readable

Author Response
Review report 4:
The manuscript, entitled 'Moving away from 12:12; the effect of different photoperiods 2 on biomass yield and cannabinoids in medicinal Cannabis', addresses issues related to optimising the photoperiod to increase the yield of desirable cannabinoids in different varieties of medicinal cannabis. This is a very up-to-date topic and I recommend its publication in Plants after taking into account the comments.
Dear Reviewer 4,
Thank you very much for your thorough review of our paper and your support in recommending its publication.
I believe I have answered all, except 1, of your requested changes. The details are contained in the text below.
Best wishes,
Sarah Purdy and Co Authors
The manuscript, entitled 'Moving away from 12:12; the effect of different photoperiods 2 on biomass yield and cannabinoids in medicinal Cannabis', addresses issues related to optimising the photoperiod to increase the yield of desirable cannabinoids in different varieties of medicinal cannabis. This is a very up-to-date topic and I recommend its publication in Plants after taking into account the comments.
Row 75 leave one dot mark – Agreed and changed
Row 87 long-day, please add (LD) Agreed and changed
Row 134 please provide the mineral content of the substrate , pH , EC salt concentration and the full composition of the nutrient solution for fertigation of plants(macro and micro nutrients what was the EC and pH )
The IP for the nutrient mix is owned by Cann Group Ltd and we cannot publish the specifics for reasons of confidentiality. The mix is based off a standard commercial blend, but we do make it ourselves. I am confident that anyone wishing to replicate our findings would be able to do using a standard, commercially available nutrient mix.
The EC and pH has now been included.
The substrate is a commercially available product (make and manufacturer supplied in text) and they state that the EC=<0.5, that is all the details provided about the mineral content, and I have now included this in the text.
Row 139 here it is necessary to explain in more detail that 10 plants from each of the three combinations where 15 plants were growing were changed the photoperiod, lengthening or shortening the photoperiod for 1/2 of them. Thus, nine combinations of 5 plants in each were obtained. Agreed and changed
Row 226 the legend posted below the chart is not clear to me, the first three line markings and color should be for 10,12 and 14L and the next 6 should have other colors or markers that would clearly characterize the 2 types of changes,
Agreed and changed
Please make changes to the combination labeling on the chart
Row 305-318 please sign on each axis of the graph for one variety and the other % THC as it is signed for the Cannatonic variety %CBDv Agreed and changed
Row 404 what do the letters a and b mean on the chart. It is a significant difference (stated in the legend)
Row 404 please correct on axes signature g.Plant-1 Agreed and changed
If surveys were conducted twice, please provide the survey dates and are the results an average of the two study dates?Row 23 proszÄ™ wymienić wszystkie kombinacje ich skróty, nazwy np. jak w tabeli 1. I think this is for a different review?!
Row 336 the line on the graph is yellow, not orange Agreed and changed
Row 274 please, either sign the axis of the graph the same as the title of the graph - flower yield or flower biomass and give the correct unit on the axis (g DW.plant-1) and do not repeat it in the title of the graph (note the same for figure1,2 and 4), or the unit always give in the title of the graph and on the axis give only the name of the studied attribute
Row 430 figure 6. I propose to leave only the abbreviation of the compound on the axis of the graph, i.e. THC, THC and CBD, and in the title of the graph write the unit - Total Yield (g.Plant-1), it will be more clearly readable
All figures changed as per your suggestion
Round 2
Reviewer 1 Report
Report on plants-2189056-R3-0207
I am pleased to say that the authors have answered my queries and made suitable changes to their earlier version of the manuscript.
Not wishing to be pedantic, but I don’t believe [line 24] that stating that day 28 was midway through flowering, for it varied between varieties. Line 141 doesn’t mention anything about halfway through flowering, so why in the abstract?
Line 26 I don’t think Dry Weight needs to be capitalised. Nor Plant in line 355.
Line 452 care with the spacing.
In the references I note that some article titles are capitalised [e.g., line 611-612] and others are not [line 613-614], so standardise please.
Reviewer 2 Report
The revised version of the manuscript may be accepted/published in Plants journal.
Author Response

(The authors gave the same response as above.)

Reviewer 3 Report
Dear authors,
Thank you for all the clarifications.
There are no further comments.
Author Response

(The authors gave the same response as above.)

Reviewer 4 Report
I accept all completed revisions and recommend the manuscript for publication
Author Response

(The authors gave the same response as above.)
